# The Epidemiology of Melioidosis and Its Association with Diabetes Mellitus: A Systematic Review and Meta-Analysis

**DOI:** 10.3390/pathogens11020149

**Published:** 2022-01-25

**Authors:** Sukanta Chowdhury, Lovely Barai, Samira Rahat Afroze, Probir Kumar Ghosh, Farhana Afroz, Habibur Rahman, Sumon Ghosh, Muhammad Belal Hossain, Mohammed Ziaur Rahman, Pritimoy Das, Muhammad Abdur Rahim

**Affiliations:** 1International Centre for Diarrhoeal Disease Research, Bangladesh (icddr,b), Dhaka 1212, Bangladesh; probir@icddrb.org (P.K.G.); rahmanhabibur803.hr@gmail.com (H.R.); sumon.ghosh@icddrb.org (S.G.); mzrahman@icddrb.org (M.Z.R.); 2Bangladesh Institute of Research and Rehabilitation in Diabetes, Endocrine and Metabolic Disorders (BIRDEM) General Hospital, Dhaka 1000, Bangladesh; barai_lovely@yahoo.com (L.B.); srafroze6@gmail.com (S.R.A.); lubna0408@gmail.com (F.A.); muradrahim23@yahoo.com (M.A.R.); 3Department of Ecology & Evolutionary Biology, The University of Tennessee, Knoxville, TN 37996, USA; belal.vetmed@gmail.com; 4School of Health, Federation University Australia, Ballarat, VIC 3353, Australia; pritimoydas@gmail.com

**Keywords:** melioidosis, diabetes mellitus, humans, systematic review, meta-analysis

## Abstract

Melioidosis is an under-recognized fatal disease in humans, caused by the Gram-negative bacterium *Burkholderia pseudomallei*. Globally, more than 35,000 human melioidosis cases have been reported since 1911. Soil acts as the natural reservoir of *B. pseudomallei*. Humans may become infected *by* this pathogen through direct contact with contaminated soil and/or water. Melioidosis commonly occurs in patients with diabetes mellitus, who increase the occurrence of melioidosis in a population. We carried out a systematic review and meta-analysis to investigate to what extent diabetes mellitus affects the patient in getting melioidosis. We selected 39 articles for meta-analysis. This extensive review also provided the latest updates on the global distribution, clinical manifestation, preexisting underlying diseases, and risk factors of melioidosis. Diabetes mellitus was identified as the predominant predisposing factor for melioidosis in humans. The overall proportion of melioidosis cases having diabetes was 45.68% (95% CI: 44.8–46.57, *p* < 0.001). Patients with diabetes mellitus were three times more likely to develop melioidosis than patients with no diabetes (RR 3.40, 95% CI: 2.92–3.87, *p* < 0.001). The other potential risk factors included old age, exposure to soil and water, preexisting underlying diseases (chronic kidney disease, lung disease, heart disease, and thalassemia), and agricultural activities. Evidence-based clinical practice guidelines for melioidosis in patients with diabetes mellitus may be developed and shared with healthcare professionals of melioidosis endemic countries to reduce morbidity.

## 1. Introduction

Melioidosis is an infectious disease in humans caused by the soil-borne, Gram-negative, facultative, intracellular bacterium *Burkholderia pseudomallei* [1]. Melioidosis is endemic in Southeast Asia and Australia [1]. Soil is the natural reservoir of *B. pseudomallei.* The organism was isolated from soil and water from many countries including, Thailand, Malaysia, Australia, China, and Bangladesh [1,2]. The first few cases of melioidosis were reported from Myanmar in 1911 [3]. As of 19 July 2020, more than 35,000 human cases were reported globally [4]. The estimated incidence rate varies from country to country; 19.6 per 100,000 person-years in Australia and 12.7 per 100,000 person-years in Thailand [5].

Transmission of this organism occurs mainly by contact with contaminated soil and water through penetrating wounds or skin abrasions, and ulcers or burns [1]. Transmission through inhalation was also reported [6,7]. Farmers working in paddy fields are frequently exposed to *B. pseudomallei*-contaminated soil and water through skin inoculation [1,5,8]. The majority of the human cases were identified during the rainy season [5,9,10,11,12]. People of all age groups are susceptible, but adults with underlying conditions were mostly affected [5,13,14,15,16]. The major clinical presentations include localized abscesses, pneumonia, and acute septicaemia [17,18,19,20,21,22,23,24]. The case fatality rate (CFR) may reach up to 40% in untreated patients [17]. Many antibiotics, including ceftazidime, carbapenems (imipenem, meropenem), piperacillin, amoxicillin–clavulanate, ceftriaxone, and cefotaxime, showed various degrees of bactericidal activity [25,26,27,28]. Parenteral ceftazidime and meropenem are the choice of antibiotics to treat melioidosis and the duration of treatment for intensive therapy was recommended to be at least 14 days. The minimum treatment durations for melioidosis with central nervous system infection, osteomyelitis, and deep-seated abscess were eight, six and four weeks, respectively [25,26]. To prevent recrudescence or relapse of melioidosis, subsequent eradication therapy has been suggested. Oral trimethoprim-sulfamethoxazole was preferred antibiotic for the subsequent eradication therapy and the duration of treatment was three to six months [25,26]. Parenteral amoxicillin-clavulanate was found effective for initial treatment, but not for severe melioidosis patients [25,29].

Reports from different countries showed that melioidosis was commonly detected in patients with diabetes mellitus [11,30,31,32]. In 2017, more than 450 million people suffered from diabetes mellitus globally, with approximately 5 million deaths [33]. People with diabetes mellitus increase the occurrence of melioidosis in a population [30]. It is still unclear why diabetic patients are at high risk of acquiring melioidosis, and hence, associated with more adverse outcomes and early death among these patients. Many review articles have been published to date on melioidosis, but no review article provided comprehensive information about the percentage of diabetics among melioidosis patients in a global perspective. The primary objective of this systematic review was to collect available information about the global distribution of melioidosis, clinical manifestation, microbiological and immunological characteristics, and risk factors of melioidosis in humans. Finally, we performed a meta-analysis to identify the relationship between melioidosis and diabetes mellitus.

## 2. Methods

### 2.1. Search Strategy and Selection Criteria

We performed a web-based in-depth literature search using four electronic databases: PubMed, PubMed Central, MEDLINE, and Google Scholar to identify full-text research articles, abstracts, case reports, and other relevant documents. Specific keywords “melioidosis” or “diabetes and melioidosis” or “*Burkholderia pseudomallei*” or “diabetes mellitus and *Burkholderia pseudomallei*” or “risk factors of melioidosis” were used to search relevant articles. We restricted our search to literature published in English only. All databases were searched from January to October 2021. We considered original articles, review articles, abstracts, and other relevant published documents. Publications from any country were included. Documents having one or more of the following criteria were excluded: (a) not relevant to the study question; (b) inadequate information; (c) duplicate data.

### 2.2. Documents Selection for This Systematic Review and Meta-Analysis

The extensive literature search revealed a total of 3156 publications. All selected documents were published between 1 January 1911 and 12 November 2021. Among these publications, we identified only 163 relevant documents that met the inclusion criteria (primary research studies). After initial screening, we performed a detailed review of selected published documents for data extraction that were relevant to our study. A total of 118 publications were identified for the full review (Figure 1). The first author thoroughly reviewed all selected articles, abstracts, and published documents for this systematic review. Finally, all extracted data were reviewed, organized, and revised based on feedback from all authors.

To examine the association between melioidosis and diabetes mellitus, we selected 39 articles for meta-analysis using the following inclusion criteria: (a) the studies were published in peer-reviewed journals; (b) the studies reported melioidosis with diabetes mellitus status; (c) the studies reported at least five melioidosis cases having diabetes mellitus. We summarized the proportions of melioidosis having diabetes from the selected articles and logit transformed before calculating inverse variance-weighted averages described by Lipsey and Wilson [34]. The meta-analysis was performed for risk ratio (RR) with 95% confidence intervals to assess the overall association between melioidosis and diabetes mellitus. We used Python 3.6 version software to analyze the metadata and produce a global distribution map [35].

## 3. Results and Discussion

Globally, more than 50 countries reported at least one human case with *B. pseudomallei* infection (Figure 2, Appendix A) [4]. More than 35,000 human cases have been reported from 1911 to 2020 (Appendix A). Diabetes mellitus was identified by many studies as the predominant underlying risk factor associated with melioidosis in humans. This study provided the latest updates on the global distribution, clinical manifestation, microbiological and immunological characteristics, preexisting underlying diseases, and risk factors of melioidosis. The meta-analysis combined data from several studies throughout the world to identify the association between melioidosis and diabetes mellitus.

### 3.1. Geographical Distribution of Melioidosis

Melioidosis is endemic in tropical countries such as Southeast Asia, Northern Australia, India, Taiwan, and China (Figure 3). The distribution of human cases mainly depends on the exposure to *B. pseudomallei* in contaminated environments [36]. The CFR varied between countries (8–60%) and the overall CFR was around 40% [37]. The highest number of reported human cases (*n* = 27,375) was in Thailand [4]. Most cases were detected from the North-Eastern provinces of Thailand and sepsis-related CFR was 40% [17,30]. Singapore reported more than 1800 cases with 16–48% CFR [4,11,21,38,39]. Malay and Indian populations living in Singapore were affected more than other groups of people [21]. The first human case was reported in Australia in 1950. So far, more than 1500 human cases have been reported and most patients were from the Northern Australia [1,4,40]. The reported CFR was 12% (*n* = 133) during 1989–2019 in Australia [40]. Cambodia, Laos and Malaysia reported more than 1000 cases of melioidosis in humans separately [4]. A hospital-based study from Cambodia reported 16.8% CFR among children with melioidosis [41]. India reported more than 500 cases and most of the cases were identified from the South-Western coastal Karnataka and northeastern Tamil Nadu [4,42,43]. A hospital-based observational study reported 23% CFR in India [44]. Sri Lanka and Bangladesh reported more than 200 and 89 cases (including some unpublished cases; personal communication), respectively [4,45]. Other Southeast Asian countries outside of Thailand, such as Vietnam, Indonesia, Philippines, Brunei, and Myanmar also reported a significant number of melioidosis cases [4]. Taiwan, China, and Hong Kong reported 318, 91, 20 cases, respectively [4]. Although many other countries reported a number of *B. pseudomallei* infection in humans, the actual burden of this disease has not been properly recognized or was underestimated in most of the countries, because of poor diagnostic facilities, microbiological culture before the administration of antibiotics, and inadequate awareness among clinicians and microbiologists [4]. Countrywide instituting improved diagnostic capabilities with trained laboratory personals and surveillance can be helpful to detect melioidosis cases accurately.

### 3.2. Clinical Presentations

The clinical presentation of melioidosis reported by many countries was diverse. Septic shock was identified as the most fatal clinical presentation of melioidosis globally [46]. The other common clinical manifestations for melioidosis were fever, pneumonia, and abscess (Table 1). Multiple organs including, lung, liver, spleen, skeletal muscle, and prostate were mostly affected [17]. Other reported clinical manifestations were genitourinary infections, prostatic abscesses, splenic abscess, liver abscess, septic arthritis, hepatomegaly, splenomegaly, abdominal pain and joint pain were mostly commonly reported in Australia [13,15,16,17,19,20,42,45,47,48].

The pattern of clinical manifestations was similar in children and adults. Children infected with *B. pseudomallei* in Thailand showed localized infections and pneumonia [64]. Children less than 15 years old were often diagnosed with cutaneous melioidosis in Australia [49]. Acute suppurative parotitis, pneumonia, and superficial soft-tissue abscess were the common clinical presentation in children in Cambodia, where parotitis is thought to be associated with ingestion of contaminated drinking water [54,55,66]. In Brazil, most of the children infected with *B. pseudomallei* had sepsis (90%) and pneumonia (80%) [67].

A number of melioidosis cases from Thailand, Australia and Laos showed recurrent infection [68,69,70,71,72]. In Thailand, 75% of the episodes among 116 patients showed relapse (infection with same strain) and 25% of the episodes were due to re-infection (infection with a new strain) [68]. Among the recurrent cases in Thailand, localized infection was the most common manifestation followed by a bacteremic, disseminated and multifocal presentation [73]. In Australia, 39 (6%) cases among the 679 survival patients were diagnosed with recurrent melioidosis during 1989–2012 [69].

### 3.3. Microbiological and Immunological Characteristics

*B. pseudomallei* is a motile, Gram-negative, oxidase positive rod mostly found in wet soils. *B. pseudomallei* has the ability to enter, survive, and replicate within host cells [74]. The cell surface of *B. pseudomallei* contains lipopolysaccharide (LPS), capsular polysaccharides and flagella that act as virulence factors [46]. Under aerobic conditions, the bacteria grow readily on routine culture media (blood and MacConkey agar) [75]. Ashdown’s agar is used as selective media for improved isolation of *B. pseudomallei* from non-sterile sites [76]. *B. pseudomallei* is able to survive; absence of nutrients; an acidic environment (pH < 4.5); wide temperature range (24–32 °C); low water content soil [1]. The role of cellular adaptive immunity to *B. pseudomallei* has been reported. *B. pseudomallei*-specific CD4+ IFN-γT-cells and CD8+ IFN-γT-cells responses were depressed in fatal cases compared to melioidosis survivors [77,78]. *B. pseudomallei* also produces humoral immunity at any stage of infection, including subclinical infection. Immunoglobulin G (IgG), IgA, and IgM were the common isotypes. Among the IgG isotypes, IgG1 and IgG2 subclasses were identified predominantly. Repeated exposure to *B. pseudomallei* produced a high-level antibody titre that might protect against severe infection [79,80,81]. A study identified a high level of *B. pseudomallei*–specific IgG2 among melioidosis survivors compared to non-survivors [82]. Focal or diffuse acute necrotizing inflammation with varying numbers of neutrophils, macrophages lymphocytes, and “giant cells” were observed in autopsy cases [83]. *B. pseudomallei* appeared to be resistant to serum bactericidal components. *B. pseudomallei* can survive and multiply within phagocytes, including macrophage/monocyte and neutrophil [84,85]. *B. pseudomallei* isolates were found sensitive to ceftazidime, cotrimoxazole, carbapenems (imipenem, meropenem), doxycycline and amoxicillin–clavulanate and resistance to ampicillin, aminoglycoside, ciprofloxacin, cefotaxime, and colistin [10,25,44,57,86,87].

### 3.4. Risk Factors Associated with Melioidosis

Patients with certain medical conditions including diabetes mellitus, thalassemia, renal disease, chronic lung disease, and malignancy increase the risk of melioidosis (Table 2) [1,19,30,31,72,88,89,90]. Among the underlying diseases, diabetes mellitus was the main underlying risk factor for developing melioidosis, which was predominant in many countries where melioidosis was endemic (Figure 4, Appendix A). Agriculture farmers working mainly in paddy fields are considered a high-risk group because they are exposed to soil and water that could contain *B. pseudomallei* [8,89,91,92].

In China, 50% of melioidosis cases were reported among rural farmers [12]. In India, 37% of melioidosis patients had a history of soil exposure [95]. Rice farmers, gardeners, and planters were more likely to be infected with *B. pseudomallei* in Thailand [19,20]. In China, farming and freshwater fishing were mainly associated with melioidosis [16]. In Bangladesh, 36% of patients with melioidosis were agriculture farmers, and 88% of the cases had a history of soil contact or environmental exposure [45]. Most of the cases (77%) from Cambodia had a history of close contact with soil and water [89].

Melioidosis was seasonal in the endemic countries. The majority of the melioidosis cases were detected during or shortly after the rainy season, which is consistent in Thailand, Australia, Singapore, India, China, Cambodia, and Bangladesh [8,9,10,11,13,91]. Moist clay soils favor the growth of *B. pseudomallei* [98]. It was perceived that farmers were more exposed to *B. pseudomallei* contaminated soil and flood water while working in the paddy fields during the monsoon [46]. Excessive alcohol consumption was observed to increase the risk of contracting melioidosis in Australia, Thailand, Cambodia, and India [15,31,40,89,90,99].

People of any age can be infected with *B. pseudomallei,* but the disease mostly occurred between 40–60 years [5]. The median age was 49 years in Thailand, 45 years in Bangladesh [30,45]. The mean age was 46 years in India, 47 years in Australia, 48 years in China, 49 years in Cambodia, 65 years in Taiwan, and 51 years in Singapore [13,14,15,16,21,42]. Melioidosis infections were also reported in children. In Cambodia, a hospital-based study reported 173 children with melioidosis from January 2009 to December 2013 and the median age was 5.7 years (range: 8 days–15.9 years). The majority of the children cases were detected during the rainy season and 76% of them had localized infections [41]. Males were more likely to have melioidosis than females. The majority of the cases were male in India (92%), Singapore (84%), China (84%), Bangladesh (80%), Taiwan (75%), Australia (75%), Thailand (69%), and Cambodia (59%) [13,14,16,19,21,42,45].The gender differences in melioidosis prevalence particularly in adult people could occur due to frequent exposure to contaminated soil and water during agriculture.

Multiple studies reported recurrent melioidosis in Thailand, Australia, and Laos [69,71,100]. Two studies showed recurrent melioidosis infections in Thailand were associated with short duration (8–12 weeks or less) of oral antimicrobial therapy, whereas another study revealed oral antimicrobial therapy with 12–16 weeks was associated with the 90% decreased risk of relapse compared with less than 8 weeks therapy [73,100,101].

### 3.5. Diabetes Mellitus and Melioidosis

Diabetes mellitus was identified as the major underlying disease for *B. pseudomallei* infection in India (81.6%), Singapore (59.3%), Malaysia (57%), Australia (37%), and Thailand (30%) [15,22,30,42,102]. Among the recurrent melioidosis cases, diabetes mellitus was also the most commonly identified underlying condition (57%) [73].

The meta-analysis showed that the overall proportion of melioidosis patients having diabetes was 45.68% (95% CI: 44.8–46.57, *p* < 0.001) (Figure 4). The range of proportions among melioidosis patients having diabetes in different studies varied from 18% to 100%. There was a significant heterogeneity across the studies (I-squared = 96.7%, *p* < 0.001). All studies included in the meta-analysis were hospital-based. Melioidosis cases were detected in both medical and intensive-care wards. Most melioidosis cases were diagnosed at various government tertiary hospitals, general hospitals, municipal hospitals, university teaching hospitals, private hospitals, and clinical laboratories. A small number of cases were identified at diabetic hospitals.

The forest plot generated from six studies showed that patients with diabetes were three times more likely to develop melioidosis than patients with no diabetes (RR 3.40, 95% CI: 2.92–3.87, I-squared= 98.2%, *p* < 0.001) (Figure 5). The range of RRs in six different studies varied from 1.5 (95% CI: 1.04, 2.10) to 13.1 (95% CI: 9.4, 18.1) (Appendix A).

The occurrence of melioidosis with diabetes mellitus could be associated with the defective innate immunity of diabetic patients and poor glycaemic control. Acute melioidosis cases with diabetes mellitus showed depressed cellular adaptive immune response compared to melioidosis cases without diabetes mellitus. The mean IFN-γ spot-forming cells (SFC) per million PBMC (Peripheral blood mononuclear cells) was 101 in acute melioidosis cases with diabetes. In contrast, the mean IFN-γ SFC was 198 in acute melioidosis cases with no diabetes [77]. Some studies reported the relationship between poor glycaemic control and infectious diseases [103]. The significant association between higher hemoglobin A_1c_ (HbA_1c_) and increased infection was identified by a few studies [104,105]. Older patients with diabetes mellitus are at risk of getting an infection than younger group of people [106]. Better glycaemic control and better treatment compliance, particularly in melioidosis endemic countries, might reduce the infection.

Patients with diabetes mellitus are at increased risk for developing infections and sepsis [107]. Hyperglycaemia in diabetes is thought to cause abnormalities of the host immune response, particularly in neutrophil chemotaxis, adhesion, and intracellular killing. The polymorphonuclear neutrophil defects driven by diabetes mellitus may be responsible for increasing in susceptibility to melioidosis. Diabetic patients are thought to be less able to kill or inactivate *B. pseudomallei* that may be triggered due to impaired phagocytosis of *B. pseudomallei*, reduced migration in response to interleukin-8, and an inability to delay apoptosis/necrosis [108]. Phagocytosis of *B. pseudomallei* was significantly impaired among diabetic patients with very poor glycemic control (HbA_1c_ level, >8.5%). The increased level of erythrocyte sedimentation rate (ESR), C-reactive protein (CRP), and leukocyte numbers were also observed in poorly controlled type 2 diabetic patients (Table 3) [52,109]. The presence of hyperglycemia in diabetes mellitus patients could play an important role for impaired phagocytic activity of neutrophils [110]. The increasing prevalence of diabetes mellitus with poor glycaemic control in melioidosis endemic countries might increase the burden of melioidosis over time.

### 3.6. Other Comorbid Conditions and Melioidosis

Renal disease was the second most underlying risk factor for melioidosis. Chronic lung disease and renal disease were reported in Australia [15,31,90]. In Thailand, melioidosis cases had a history of chronic renal failure, hematologic diseases, connective tissue diseases, chronic liver diseases, and tuberculosis [20,82]. In Singapore, 35.5% of melioidosis had hypertension and 15.3% had renal impairment (15.3%) [21]. Another study from Singapore reported that melioidosis patients had other underlying conditions such as tuberculosis, asthma, chronic obstructive pulmonary disease, and neoplasm [38]. In Cambodia, 1% of melioidosis patients had tuberculosis [53]. In India, 3.5% of melioidosis patients had chronic kidney disease and 2.6% had sickle cell disease [42]. In Malaysia, chronic renal failure and chronic lung disease associated with melioidosis were reported [22]. Thalassemia is identified as a significant risk factor for melioidosis in children in Malaysia [96]. In Taiwan, the majority of the melioidosis patients (90%) had underlying conditions; chronic renal disease (20%), malignancies (13%), hypertension (37%), coronary heart disease (27%), cerebrovascular disease (20%), and liver cirrhosis (13%) [48]. Comorbidities were less commonly reported in children in Cambodia. Only a few children had thalassemia, chronic renal failure, probable acute lymphoblastic leukemia, and systemic lupus erythematosus [41].

## 4. Conclusions and Recommendations

This meta-analysis highlights a high prevalence of diabetes in patients with melioidosis and a significant association between melioidosis and diabetes mellitus. The poor glycaemic control of patients with diabetes mellitus raises significant concern about the morbidity of melioidosis. Our findings suggest that agriculture farmers having diabetes mellitus in melioidosis endemic areas should avoid direct contact with soil and water during the rainy season. Better glycaemic control might reduce the morbidity of melioidosis. More research works are needed to design and evaluate both pharmaceutical and behavior change interventions. Early treatment with appropriate antibiotics after its detection is crucial in improving patient outcomes and minimizing the severity. Evidence-based clinical practice and diagnostic guidelines for suspected melioidosis cases with diabetes mellitus should be urgently developed and shared with healthcare professionals and laboratory personnel in melioidosis endemic countries to reduce fatality.

## Figures and Tables

**Figure 1 pathogens-11-00149-f001:**
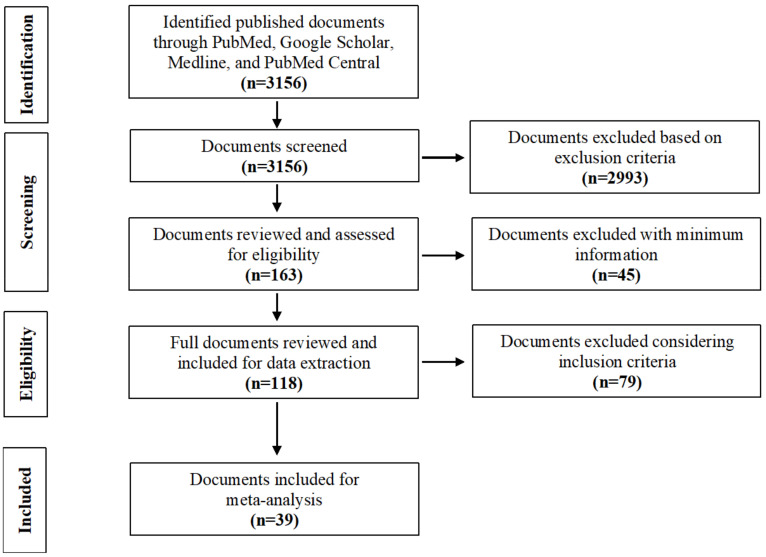
Flow diagram of the document search and selection.

**Figure 2 pathogens-11-00149-f002:**
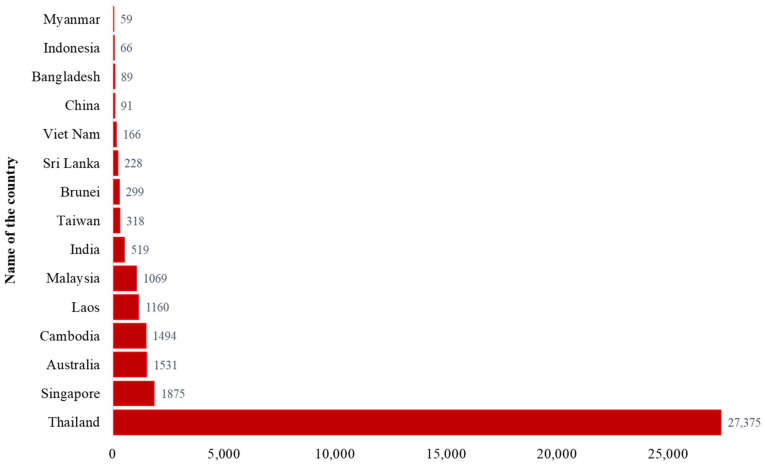
Top 15 countries reported more than 50 human melioidosis cases, 1911–2020 (Data source: Appendix A. Country data summary. Microbiology Department at Mahidol Oxford Tropical Medicine Research Unit. https://www.melioidosis.info/info.aspx?pageID=107&contentID=1070102, accessed on 16 May 2021) [4].

**Figure 3 pathogens-11-00149-f003:**
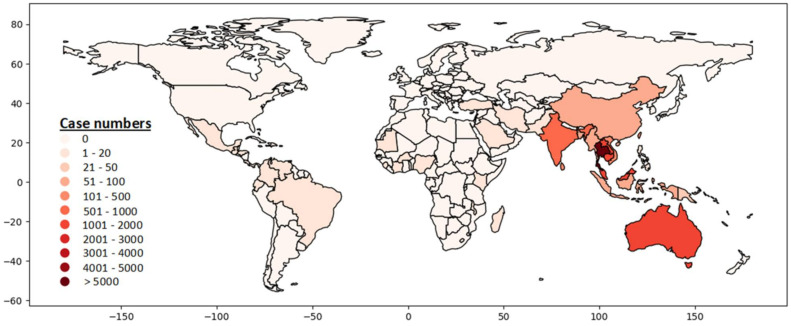
Global distribution of melioidosis in humans, 1911–2020 (Data source: (Appendix A. Country data summary. Microbiology Department at Mahidol Oxford Tropical Medicine Research Unit. https://www.melioidosis.info/info.aspx?pageID=107&contentID=1070102, accessed on 16 May 2021) [4].

**Figure 4 pathogens-11-00149-f004:**
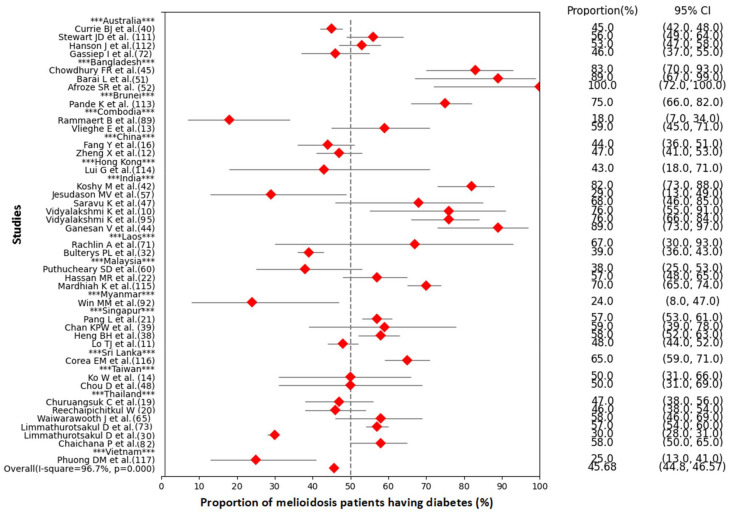
Forest plot of the meta-analysis on proportion of diabetes mellitus in melioidosis positive cases (Data source: Appendix A).

**Figure 5 pathogens-11-00149-f005:**
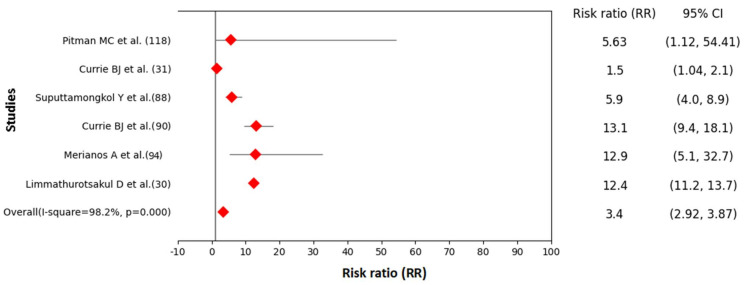
Forest plot showing association between melioidosis and diabetes mellitus in humans (Data source: Appendix A).

**Table 1 pathogens-11-00149-t001:** Country wise reported clinical presentations for melioidosis in human.

Country	Manifestation	References
Australia	Fever, pneumonia, genitourinary infections, abdominal pain, skin abscesses, osteomyelitis, septic arthritis, soft tissue abscess, and encephalomyelitis, genitourinary infection, communicating hydrocephalus and encephalomyelitis.	[15,31,40,49,50]
Bangladesh	Fever, abdominal pain, arthritis, cough, lung abscess, prostate abscess, liver abscess, skin abscess, pneumonia, septic arthritis and meningitis.	[45,51,52]
Cambodia	Fever, cough, chest pain, weight loss, acute suppurativeparotitis, pneumonia, Superficial soft-tissue abscess, Lymph-node abscess, Meningitis, Bone/joint infection, Deep abscesses, urogenital infection, shock and multi-organ failure.	[13,41,53,54,55]
China	Fever, pneumonia, septicaemia, visceral abscess, urinary tract infection, lymphadenitis, arthritis, parotitis, orchitis, soft tissue abscess and prostatic infection.	[12,16,56]
India	Fever, visceral abscess, septic arthritis, renal failure, abdominal pain, pneumonia, hepatomegaly, osteo-myelitis, splenomegalyand septicemia.	[10,42,47,57]
Laos	Fever, weight loss, productive cough, acute bilateral supraclavicular lymphadenitis, septic arthritis and spleen abscess.	[58,59]
Malaysia	Fever, pneumonia, septicemia, shock, lung abscess, cervical abscess, submandibular abscess, axillary abscess, skin abscess, muscle abscess, liver abscess and brain abscess.	[18,22,60,61,62]
Singapore	Fever, pneumonia, acute respiratory distress syndrome (ARDS), abscess, abdominal pain, vomiting, diarrhea, dysuria and haematuria.	[21,38,39]
Taiwan	Fever, cough, pneumonia, abdominal pain, septicemia, soft-tissue abscess, mycotic aneurysm and renal failure.	[14,48,63]
Thailand	Fever, pneumonia, acute respiratory distress syndrome (ARDS), splenic abscess, liver abscess, muscle abscess, prostatic abscesses, renal abscess, parotid gland abscess, submandibular node abscess, septic arthritis, osteomyelitisand facial cellulitis.	[19,20,64,65]

**Table 2 pathogens-11-00149-t002:** Country wise identified significant risk factors for melioidosis in humans.

Country	Risk Factors	References
Australia	Diabetes mellitus	[15,31,90,93,94]
	Exposure to soil and water	[49]
	Alcoholism	[31,94]
	Old age	[90]
	Chronic lung disease	[31,90]
	Chronic renal disease	[90]
	Heart disease	[31]
	Rainfall	[9]
Bangladesh	Diabetes mellitus	[45]
Cambodia	Inappropriate antibiotic therapy	[13]
	Close contact with wet soil	[89]
	Underlying chronic disease	[89]
India	Diabetes mellitus	[10,42,95]
	Alcoholism	[47]
	Rainfall	[10]
	Old age	[95]
Malaysia	Diabetes mellitus	[22]
	Thalassemia (children)	[96]
Singapore	Diabetes mellitus	[11,38]
	Old age	[11,38]
Thailand	Diabetes mellitus	[30,88]
	Thalassaemia	[88]
	Lung disease	[19]
	Inappropriate antibiotic therapy	[19,65]
	Working in rice field	[97]
	Exposure to rain	[97]
	Exposure to soil and water	[88]
	Old age	[30]

**Table 3 pathogens-11-00149-t003:** Biochemical and hematogical characteristics of the poorly controlled glycemia, well-controlled glycemia and non-diabetic patients [109].

Characteristics	Poorly-Controlled Glycemia	Well-Controlled Glycemia	Non-Diabetic
Glycated hemoglobin A1c (HbA1c) (%)	10.1 ± 1.2	6.1 ± 0.9	5.1 ± 0.4
Erythrocyte sedimentation rate (ESR) (mm/h)	32.8 ± 18.4	42.1 ± 24	10.7 ± 7.6
C-reactive protein (CRP) (mg/L)	8.7 ± 7.3	7.0 ± 5.2	2.0 ± 0.9
Red blood cell count (RBCC) (10^12^/L)	4.9 ± 0.6	4.6 ± 0.3	4.8 ± 0.5
White blood cell count (WBCC) (10^9^/L)	8.05 ± 2.5	6.4 ± 1.4	5.6 ± 1.1
Lymphocytes (10^9^/L)	2.25 ± 0.39	1.75 ± 0.41	1.84 ± 0.48
Monocytes (10^9^/L)	0.59 ± 0.21	0.54 ± 0.14	0.45 ± 0.11
Neutrophils (10^9^/L)	4.86 ± 2.2	3.84 ± 1.13	3.04 ± 0.72
Platelets (10^9^/L)	270.8 ± 91.7	241.8 ± 117.7	236.5 ± 47.7

## Data Availability

All data are available from the corresponding author after reasonable request.

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
