# Peer review of "The Epidemiology of Melioidosis and Its Association with Diabetes Mellitus: A Systematic Review and Meta-Analysis"

_pathogens, 2022, doi:10.3390/pathogens11020149_

Round 1

Reviewer 1 Report

Diabetes is the most common risk factor for melioidosis. However, there are some evidences showing that diabetes was significantly associated with lower mortality than non-diabetes. I do not agree with the conclusion in lines 26-28 and line 316 and other related sentences inthe manuscript that “Better glycaemic control might reduce the severity of melioidosis.” In my view, better glycaemic control may reduce morbidity but not mortality from melioidosis.

Line 55-57: Ceftazidime and meropenem are the choice of treatment melioidosis for both severe and non-severe cases. I would suggest to remove “severe” from the sentence in line 55.

Line 110-11: Figure 2 and Supplement table 1 do not show the total 35,000 human cases. Please add the total count in the table and/or figure.

Line 124: The highest number of reported human cases (n=20,346) was in Thailand. The number of cases in the text do not match with the data in Figure 2 and Supplemental Table 1(N = 27,375).

Line 178: Please change from “Ashdown’s” to “Ashdown agar”.

Line 189-190: The data did not demonstrate “This B. pseudomallei-specific IgG2 could play an important role in reducing CFR”. I would suggest to remove this sentence.

Line 192: Resistance to serum bactericidal components may not be the only mechanism for intracellular multiplication and survival of B. pseudomallei. I would suggest to separate these two sentences and remove “suggesting”.

Line 201: Data in Table 2 does not show about severity. Please modify the sentence.

Line 225: Please remove “infections” after Melioidosis.

Line235-238: I am confused with the sentence “Two studies showed recurrent melioidosis infections in Thailand….the lowest hazard ratio”.

Line 263-268: The mean SFC of acute melioidois with diabetic melioidosis patients was lower or higher than non-diabetic cases. Please check the data.

Author Response

Comment: Diabetes is the most common risk factor for melioidosis. However, there are some evidences showing that diabetes was significantly associated with lower mortality than non-diabetes. I do not agree with the conclusion in lines 26-28 and line 316 and other related sentences inthe manuscript that “Better glycaemic control might reduce the severity of melioidosis.” In my view, better glycaemic control may reduce morbidity but not mortality from melioidosis.

Response: Thanks for your suggestions. We have revised this section considering your suggestion (page no. 1 & 11, line no. 28, 321 & 323).

Comment: Line 55-57: Ceftazidime and meropenem are the choice of treatment melioidosis for both severe and non-severe cases. I would suggest to remove “severe” from the sentence in line 55.

Response: Thanks for your suggestion. We have removed “severe” from the revised manuscript (line no. 52)

 Comment: Line 110-11: Figure 2 and Supplement table 1 do not show the total 35,000 human cases. Please add the total count in the table and/or figure.

Response: We have included total case numbers and deaths in the supplement table 1.

 Comment: Line 124: The highest number of reported human cases (n=20,346) was in Thailand. The number of cases in the text do not match with the data in Figure 2 and Supplemental Table 1(N = 27,375).

Response: Thanks. The correct number will be 27,375. We have included correct number in the revised manuscript (line no. 122)

Comment: Line 178: Please change from “Ashdown’s” to “Ashdown agar”.

Response: Thanks for your suggestion. We have added correct information (line no. 178)

Comment: Line 189-190: The data did not demonstrate “This B. pseudomallei-specific IgG2 could play an important role in reducing CFR”. I would suggest to remove this sentence.

Response: Considering your suggestion, we have removed the sentence from the revised manuscript.

Comment: Line 192: Resistance to serum bactericidal components may not be the only mechanism for intracellular multiplication and survival of B. pseudomallei. I would suggest to separate these two sentences and remove “suggesting”.

Response: Thanks. We have revised this section considering your suggestion (line no. 1923)

Comment: Line 201: Data in Table 2 does not show about severity. Please modify the sentence.

Response: We have revised the sentence (line no. 201)

Comment: Line 225: Please remove “infections” after Melioidosis.

Response: We have removed the suggested word.

Comment: Line235-238: I am confused with the sentence “Two studies showed recurrent melioidosis infections in Thailand….the lowest hazard ratio”.

Response: We have revised the sentence with appropriate information.

Comment: Line 263-268: The mean SFC of acute melioidois with diabetic melioidosis patients was lower or higher than non-diabetic cases. Please check the data.

Response: Very useful comment. We have added correct information in the revised manuscript.

Reviewer 2 Report

A rather extensive review of published literature thus far. However, the association of melioidosis and diabetes mellitus could be improved with further elaboration of underlying immune dysfunction. 

Author Response

Comment: A rather extensive review of published literature thus far. However, the association of melioidosis and diabetes mellitus could be improved with further elaboration of underlying immune dysfunction. 

Response: Thanks for your comments. We have tried to provide more information about melioidosis, diabetes mellitus and associated immune dysfunction.

Reviewer 3 Report

The meta-analysis of association between human melioidosis and DM is interesting. It would be more useful for readers if authors performed quantifying of heterogeneity including identifying bias in this meta-analysis. It is essential for authors to discuss more on selecting studies of melioidosis cases in DM clinics comparing to studies of melioidosis cases in outpatient or ICU wards.  This might be as important as patients’ country of origins.  Figure 2 and Figure 3, the information of data source (cite the publication and period of time of data collections) should be described in details.

Author Response

Comment: The meta-analysis of association between human melioidosis and DM is interesting. It would be more useful for readers if authors performed quantifying of heterogeneity including identifying bias in this meta-analysis. It is essential for authors to discuss more on selecting studies of melioidosis cases in DM clinics comparing to studies of melioidosis cases in outpatient or ICU wards.  This might be as important as patients’ country of origins.  Figure 2 and Figure 3, the information of data source (cite the publication and period of time of data collections) should be described in details.

Response: There was a significant heterogeneity across the studies (I-squared = 96.7%, p <0.001). We included this information in result section (page no. 9, line no. 258-259). All studies included in the meta-analysis were hospital-based. Most cases were diagnosed at various government tertiary hospitals and clinical laboratories. A small number of cases were identified at diabetic hospitals. We have included more information about this in the revised manuscript (page no. 9, line no. 258-263). We have included data source and duration of reporting in the revised manuscript (Figure 2 & 3, page no. 4 & 5, line no. 114&146).

Reviewer 4 Report

Please see the attached report

Author Response

Special Issue Editor

Pathogens

Subject:  Responses to the review comments

Dear Dr. Jose Muñoz Gutiérrez,

Thank you for sharing additional comments and suggestions from 4th reviewer. All comments and suggestions are very useful! We have responded to all comments and incorporated all because we believe these strengthen the manuscript. We have added more latest information and made necessary changes in the revised version of the manuscript uploaded with this submission. Detailed responses to each comment are given below for your consideration.

Sincerely,

Sukanta Chowdhury, PhD

Associate Scientist

Programme for Emerging Infections

Infectious Diseases Division

icddr,b

Email: sukanta@icddrb.org

Reviewer -4:

Comment: The authors expended a tremendous amount of effort to collect the literature on melioidosis and further down- selected references/data to perform their meta-analysis to “explore” the relationship between melioidosis and diabetes. The word “explore” is vague and should be more direct or explanatory. We interpreted their aim as more of a question, such as “investigate how does diabetes effect the subject to getting melioidosis” or “determine how much of a risk factor is diabetes in people getting melioidosis”? Much has been reported previously on this relationship (risk factor) of diabetes and melioidosis. They mentioned that they did “a scoping review” to collect the information for the meta-analysis. Because “scoping review” and “meta-analysis” was in the title of their manuscript, this effort was defined by these methods to accomplish what they wanted to do. However, it also raised the question if it really was a scoping review or could it have been a systematic review of the melioidosis literature? Also, can a meta-analysis be done on results from a scoping review or more properly on results from a systematic review? These questions caused confusion in this reviewer when reading the manuscript. It appears that they collected the information that they wanted, but it could have been presented and organized in a clearer manner.

Response: Thanks for your valuable comment. We have removed the word ‘explore’ and revised the sentence. We rethought and have agreed with you that ‘systematic review’ is more appropriate for this manuscript.

Comment: As for the differences between a scoping review or systematic review, many reports have been published to try to define the differences between these two types of reviews (Arksey and O’Malley, 2005, Internat J of Social Res Meth: Theory and Practice, 8(1),p19; Pham et al., 2014, Res Syn Meth, 5, p371; Munn et al., 2018, BMC Med Res Meth, 18,p143; Sargeant and O’Connor, 2020, Front Vet Sci, 7, 11.). We were not sure if the authors did a scoping review (or mapping study) for this manuscript although they said it was a scoping review. They also asked the question (“explore”?) about the relationship between melioidosis and diabetes mellitus. Systematic reviews are more question driven than scoping reviews (see reports mentioned above). 2

Response: Thanks for sharing information. Very useful! We have agreed with you that ‘systematic review’ is more appropriate for this manuscript. We have incorporated this in the revised manuscript.

Comment: The authors should better define the type of review they are performing, and explain why they are choosing one over the other (justification). Both these statements/questions are being raised because the type of review was emphasized in the title of the manuscript. Most reviews on melioidosis do not define the type of review that was completed. Scoping reviews also identify gaps in the existing literature. The present manuscript does not identify or report any gaps in their review. They should devote a few lines or a paragraph to the gaps they found in their scoping review (if that was the type of review they carried-out).

Response: Thanks for sharing information. Very useful! We have agreed with you that ‘systematic review’ is more appropriate for this manuscript. We have incorporated this in the revised manuscript.

Comment: The authors should be more clear on what they want to do rather than “explore”. See first paragraph above under major concerns, or remarks under number 4 below.

Response: We have removed the word ‘explore’ and revised the sentence. We rethought and have agreed with you that ‘systematic review’ is more appropriate for this manuscript.

Comment: The Abstract is a bit disorganized and sometimes confusing. Ln17, they carried out a meta-analysis and then Ln19, they performed a scoping review? Which came first?

Response: Thanks for your comment. We have revised this section considering your suggestions (line no. 17-18).

Comment: 3a. Ln15. “Melioidosis commonly occurs in patients with diabetes mellitus.” This information has been published in many previous reports. So, why are we asking about the relationship or association between melioidosis and diabetes mellitus in the present report? Actually, many reviews on melioidosis do not give the percentage of diabetics among melioidosis patients. Some do, however. The authors might want to suggest that the previous information about the relationship was incomplete or not known?

Response: We have added more information in the revised manuscript (line no. 67-69).

Comment: 3b. Ln16. Increasing prevalence of diabetics does not increase the “risk” of a population, but people with diabetes increase the incidence or occurrence of melioidosis in a population. The risk is still 45-50% of melioidosis cases. It does increase the risk of an individual who developed diabetes. Delete the word …understandably… This statement has been mentioned in more than one place and should be modified in all cases to be clear.

Response: We agree with you and revised the section (line no. 16-17).

Comment: 3c. Ln17. Statement “We carried out a meta-analysis…” before they performed a scoping review (Ln19) to collect the data?

Response: We carried out a systematic review first and then meta-analysis to investigate how does diabetes effect the subject to getting melioidosis. We have revised the section considering your suggestion (line no. 18-19).

Comment: 4. The last paragraph of the Introduction like the Abstract was awkward. They should also state the objective of the study clearly (1ST, 2ND, and 3rd objective). They asked the question “what was the relationship between melioidosis and diabetes?” See remarks above in the first paragraph under major concerns. At least that was how this reviewer interpreted the statement. 1. They performed a scoping review (or systematic review?) to collect the available information on melioidosis (and clinical presentation, patient health condition, treatment, etc). 2. They performed a meta-analysis to examine the relationship between melioidosis and diabetes in the collected data (with their elimination or exclusion limits). Could they do the meta-analysis before they collected the data? 3. They provided updated information about the global information of melioidosis, etc. It seems like some of the chronology of the way they carried out this study was not stated clearly.

Response: We carried out a systematic review first and then meta-analysis to investigate how does diabetes effect the subject to getting melioidosis. We have revised the section considering your suggestion (line no. 72-77).

Comment: 5. The manuscript could be better organized. For example the discussion on diabetes and other risk factors. Combine results on diabetes in one section, and other risk factors and comorbid conditions into another section(s). But, the diabetes should be first with Table 2 and Figs 4 and 5 because it is one of the main objectives or aims of the study.

Response: We tried to provide available information first about the global distribution of me-lioidosis, clinical manifestation, microbiological and immunological characteristics, and risk factors of melioidosis in humans. Later, we explained detail about the association between diabetes and melioidosis.

Comment: 6. The figures in general are inadequately labeled. See some remarks below for individual figures. Most do not have any legend other than a title of the figure. 3

Response: Thanks for your comment and added more information.

Comment: 1. Ln12. …the “Gram-negative” should be capitalized

Response: Corrected

Comment: 2. Ln32-33. …Gram-negative, facultative, intracellular bacterium…. This microorganism can live in soil, water, and serum and not always within cells. Hence “facultative”.

Response: Thanks for your suggestion. Added the text “facultative”.

Comment: 3. Ln 45. …ulcers, or burns (1).

Response: Revised.

Comment: 4. Ln46. ..paddy fields are frequently…

Response: Corrected.

Comment: 5. Ln51. Delete first part of sentence -Melioidosis mainly causes sepsis and. Start with The case….

Response: Corrected.

Comment: 6. Ln56. ..therapy was recommended…..

Response: Corrected.

Comment: 7. Ln57. ….treatment duration of melioidosis….

Response: Corrected

Comment: 8. Ln58. …abscess was eight,

Response: Corrected

Comment: 9. Ln60 …..was the preferred..

Response: Corrected

Comment: 10. Ln61 ..treatment was three to ….

Response: Corrected

Comment: 11. Ln67 ..delete “is certainly”..See statement 3b above.

Response: Corrected.

Comment: 2. Ln70 ..we explored the relationship..

Response: Revised

Comment: 13. Ln87 …revealed a total of ….

Response: Corrected.

Comment: 14. Ln89 ..relavent documents that met…..

Response: Corrected

Comment: 15. Ln93 …abstracts, and published documents…. Were these abstracts and published documents “peer-reviewed, as stated in Ln82, like the original article? (Some very old articles are not easy to examine or obtain, but they may have an abstract). If you used any “published documents” in your meta-analysis, did you reference them? (Like ref #4?).

Response: Thanks for your suggestion. We have revised the section (line no. 89-90).

Comment: 16. Ln97 Figure 1 legend is very brief, as well as the Figure legends for Fig. 2, 3, and 4. They should be more explanatory as to their content (and how the information was obtained?).

Response: We included the sources of information for Fig. 2, 3, and 4.

Comment: 17. Ln112 …underlying risk factor….

Response: Revised

Comment: 18. Ln113 The statement “This study explored the association between melioidosis and diabetes mellitus” has been stated more than once and this reviewer interpreted it as a question. A) the authors should state it as a question as a rationale for the study. Or state it as a hypothesis – there is an association between melioidosis and diabetes mellitus. B) it should be more explanatory, such as a risk factor compared to other risk factors, as was listed in Table 2.

Response: Thanks for your suggestions. We have revised the section (line no. 124-126).

Comment: 19. Ln118 Fig. 2 legend. It has ref.4, but does ref. 4 have the same numbers or different for the top 15 countries? It lists only 14 countries. If different, why is it different. Wasn’t it part of the reason for the review to update the number/cases of melioidosis? Up to November 2021?

Response: We included year of reporting and source of information for figure 2 (line no. 128-129).

Comment: 20. Ln124 In Fig2, Thailand, there is a number of 27375 of the no. of cases of melioidosis, but the number of cases in ref 4 was 20,346. A) Why was there a difference in these two numbers? B) Are the 4 numbers in Fig2 updated to November 2021? Maybe this should be mentioned in the Fig. legend rather than say they come from ref 4. C) Is the case fatality rate (CFR) the same for the updated numbers in Fig2 compared to ref 4? This is confusing. You could compare them and mention the increase in cases(look at objective Ln71, end of Introduction). This is true for all the Results discussed in this paragraph (3.1. Geographical distribution of melioidosis).

Response: Added correct statistics. Revised throughout the manuscript.

Comment: 21. Ln148, Fig3. What period was the data from (1911 to 2021?). Is it total , or per 100,000, or what?

Are the numbers from documents (N=39) that were finally used for the review and meta-analysis? Were these microbiologically confirmed cases or total cases but some not microbiologically confirmed?

Response: We included year of reporting and source of information for figure 2 (line no. 128-129).

Comment: 22. Ln158, Table 1. Were these clinical presentations from the final 39 down-selected studies? Counted 37 references. Are there two that were not included? Or were they incomplete? Column should be labeled “Clinical Manifestations” rather than just Manifestation.

Response: Information about clinical presentations was obtained from 118 studies.

Comment: 23. Ln159 The clinical manifestations were similar in children and adults.

Response: Revised.

Comment: 24. Ln159-160 Children infected with B. pseudomallei in Thailand….

Response: Revised.

Comment: 25. Ln161 Children less than 15 years old were often diagnosed with cutaneous melioidosis…

Response: Revised

Comment: 26. Ln163 …the common clinical presentation in children …

Response: Revised.

Comment: 27. Ln171-172 …diagnosed with recurrent

Response: Revised

Comment: 28. Section 3.3. Microbiological etc. This section seems out of place and was a chaotic mixture of characteristics. It deals with the microorganism, but the section was been placed between clinical presentation and risk factors of melioidosis. It may be better to place the section at the beginning of the Results/Discussion section, Section 3.1. , but shorter in length. Ln177 to 181 could be discarded. The section could be called “Microbiological and immunological Characteristics” which should cover some of the information that would be more pertinent to this review. Sentence covering Ln184 to 185, should be eliminated because it was not clear “humoral immunity at any stage of infection”?

Response: Thanks for your suggestions. We have revised the section.

Comment: 29. Ln201 The statement …malignancy “exaggerate melioidosis severity” was a bit awkward. This section discusses Risk factors, and risk factors make the patient more “susceptible” to melioidosis. Or do the authors mean that risk factors “enhance” or make melioidosis “more severe”?

Response: Revised

Comment: 30. Ln203 …considered a high risk group because they are exposed to soil and water that could contain B. pseudomallei (8,77,79,80).

Response: Revised

Comment: 31. Ln205 …humans.

Response: Revised

Comment: 32. Table 2. It lists Risk Factors in different countries. It could be more informative. A. There should be a column that lists the number (n=?) of patients/people in the different groups (obtained from the references cited). B. Also, what percentage of the people in each of these risk factor groups. It appears the percentage of some people in the groups from the different country are mentioned in the Results/Discussion. These should be part of the Table 2. C. Why are only these countries listed in Table 2? In Fig 2, there are 14 countries listed with more than 50 human melioidosis cases. Is there information from these that could be included in the risk factor Table 2? Was there a limitation that excluded the other countries from Table 2? 5

Response: We agree with you. Many countries reported only few cases. If we include all countries in the graph, it will not be visualized well. We included all countries information in supplementary table 1.

Comment: 33. Ln206. Melioidosis in China is mentioned, but China was not in Table 2? 34. Ln229. Male gender was discussed but gender as a risk factor (male vs females) was not listed in Table 2?

Response: In table, we only mentioned significant risk factors but we mentioned other risk factors in the narrative section.

Comment: 35. Ln240 to 249. If the purpose of Table 2 was to show that diabetes mellitus was the most frequent risk factor, this paragraph should be the first paragraph under section 3.4. Risk factors associated with melioidosis. Also, see item 32 above. This paragraph could be shorten or rewritten by eliminating redundancies, for example the first and second sentences could be combined.

Response: We have revised the section considering your suggestions.

Comment: 35. Ln234 to 238 These lines should be at the end of the section on Microbiology and Immunological Characteristics because they discussed recurrent infection and not a risk factor. Unless it was a risk factor, but it was not listed in Table 2 as a risk factor.

Response: Here, we showed that short duration (8-12 weeks or less) of oral antimicrobial therapy as risk factor for recurrent infection.

Comment: 36. Ln241 Patients with specific underlying diseases were…. (not all diseases are risk factors, like AIDS, as an example). Or Tb?

Response: We have removed non-relevant information from the revised manuscript.

Comment: 37. Ln251 Fig4, should have left column labeled at the top as Country/Source, and x-axis labeled Proportion (%). Again, the figure legend (there is none) is lacking information about the source of the data, etc. Only the title of the figure is given. Also check reference numbers to match the references. For example Gassiep (60)?

Response: Revised

Comment: 38. Ln256 plot generated from five studies… However, it looks like there are 6 studies on the plot? Also, check the reference numbers to match references listed.

Response: The forest plot is generated from six studies. Added correct information.

Comment: 39. Ln261 No figure legend, just a title. Left column should be labeled as “Studies” or Reports. Bottom axis should be labeled Risk Ratio (RR)?

Response: Revised

Comment: 40. Ln262 The occurrence of melioidosis with diabetes….

Response: Revised

Comment: 41. Ln267 ..the mean IFN-γ SFC was 198….

Response: Revised

Comment: 42. Ln289 Table 3 should be more informative. How many patients/people were in each group (poorly controlled glycemia (n=?), Well controlled glycemia (n=?), and Non-diabetic(n=?) Was there any significant differences between these groups for the different characteristics (compared with the non-diabetic group)?

Response: We used the information from other study. We included citation for detail information.

Comment: 43. Ln293 ….underlying risk factor for melioidosis.

Response: Revised

Comment: 44. Ln311 …a high prevalence of diabetes in patients with melioidosis

Response: Revised

Round 2

Reviewer 4 Report

There are still some concerns about the presentation of some of the information and organization of the report. 

Major concerns:

  1. Ln213 Table 2.

As we have noted previously, there should be more information in this table that identifies the “significant risk factor for melioidosis in humans”.  The proposed risk factors are listed by country and references.  There should be at least two more columns that includes the “n” for the number of people associated with each risk factor, and the % of melioidosis patients with the particular risk factor for each country.  As it currently stands, the Table just lists the proposed risk factors without any weight for each risk factor listed for readers to consider.  It should be more informative.  There are reviews, which we have mentioned previously, that do not include this information but just lists the risk factors.  This report should supply that information from its meta-analysis.  Diabetes mellitus was listed first in five of the seven countries included in the Table.  Why? It is not enough just to make a list (like Table 1) that does not inform the reader what is the predominant (%) risk factor reported for each country, and how the risk factors rank relative to each other.  No other Table (including the Supplementary Tables) or figure in this report shows the relationship between diabetes mellitus and other individual risk factors.  The “n” is important to show how many people are being considered in each category.

There should also be a footnote or explanation after the title as to why only these countries (7) were considered and not others.  If the authors believe only these countries/references have adequately reported the data to their standards/requirements, they should say this.

  1. Ln180 3.3. “Microbiological and immunological characteristics.”

As we have noted previously, this section is out of place between Clinical presentations and Risk factors associated with melioidosis.  This section is a wide assortment of information about B. pseudomallei, its ability to grow in cells, its surface characteristics, its ability to grow in certain media, etc. As it is, this paragraph would be difficult to fit anywhere in this particular review and does not contribute any information that would be pertinent to the subject of the review. Hence, it is not clear why this paragraph belongs in this review that is supposed to be focused on diabetes mellitus, risk factors, and melioidosis.  If this paragraph were eliminated, it would not affect the overall report. 

Other concerns.

Ln12 …Gram-negative…   -there should be a hyphen between Gram and negative in all cases through out the manuscript.

Ln17-18  …to investigate to what extent that diabetes mellitus affects the patient in getting melioidosis.

Ln67  ..published to date on melioidosis,…

Ln68-69  ..patients in a global perspective.

Ln146 …was underestimated….

Ln165 …was similar in children…

Ln167…diagnosed with cutaneous…

Ln209 …which was predominant…

Ln210 ..melioidosis was endemic..

Ln224   It was perceived that...

Ln265  ..from seven studies –not sure you mean seven rather than six

Ln267  …in six different – not sure you mean six rather than five

Ln343   The reporting of the journals in the references is mixed.

This manuscript is a resubmission of an earlier submission. The following is a list of the peer review reports and author responses from that submission.

Round 1

Reviewer 1 Report

This is a review article on Melioidosis. While it is relevant, there is little new in this article. I have the following comments.

  1. The distribution of cases will depend upon the collection of clinical samples for microbiological culture before the administration of antibiotics. It will also depend upon the ability of the laboratory to correctly identify the organism This needs to be emphasised. It should also be acknowledged that the absence or low incidence of disease may be due to the above factors.
  2. It would be useful to tabulate the mortality rates by country
  3. Section 3.2 Clinical presentation. This entire section can be summarised and referred to Table 1
  4. Section 3.5 The presence of antibodies does not provide immunity. True "immunity is cellular.
  5. The manuscript will need extensive grammatical and spelling amendments.

Reviewer 2 Report

The discussion about melioidosis and diabetes is important and the article contributes to the attention of this public health problem. Some considerations:

Line 12. Caused by Gram negative bacterium.

Lines 41 and 42. Transmission of melioidosis: iInfection results from exposure through skin inoculation, inhalation and ingestion of Burkholderia pseudomallei. Relevant references were not cited.

Line 52. Treatment of melioidosis (duration of treatment). Relevant and recent references were not cited.

Reviewer 3 Report

The association of melioidosis and  Diabetes Mellitus has been a topic for debates for many decades. This paper showed the results of meta-analysis for the relationship between melioidosis and diabetes and the information of other risk factors. This is a good review. I have some comments.

Line 2 and 250: The proportion of melioidosis cases should be in percent (43.53%).

Line 87 and Figure 1: Was the data for the full review 98 or 105 publications? The number from text and flow diagram are not the same.

Line 169-174: The data presented here is old. Please add more recent  information for the recurrent and relapse cases for melioidosis.

Line 186: Please correct "B. pseudomallei".